# The Relationship of Self-Reported Physical Activity Level and Self-Efficacy in Physiotherapy Students: A Cross-Sectional Study

**DOI:** 10.3390/ijerph22071029

**Published:** 2025-06-27

**Authors:** Lāsma Spundiņa, Una Veseta, Agita Ābele

**Affiliations:** 1Department of Health Psychology and Pedagogy, Riga Stradiņš University, LV-1007 Rīga, Latvia; una.veseta@rsu.lv; 2Department of Sport and Training Theory, Pedagogy, Psychology and Pedagogical Internship, RSU Latvian Academy of Sports Education, LV-1007 Rīga, Latvia; agita.abele@rsu.lv

**Keywords:** self-efficacy, physical activity, physiotherapy students

## Abstract

Physical activity plays a critical role in health and well-being, particularly during students’ academic development. This study explores the relationship between self-efficacy and physical activity among physiotherapy students, recognizing self-efficacy as a key factor influencing exercise behavior. Despite awareness of physical activity’s benefits, academic demands may hinder participation, reducing confidence in maintaining an active lifestyle. A total of 244 physiotherapy students (mean age 24.44 ± 7.56 years) completed the General Self-Efficacy Scale (GSES) and the International Physical Activity Questionnaire—Short Form (IPAQ-SF). The results showed that the self-efficacy scores ranged from 17 to 40, with a mean of 30.44 (±3.93), indicating moderate to high levels. In terms of activity, 40.3% of students reported sufficient activity (high level), 51.7% reported moderate activity (meeting minimum guidelines), and 8.05% reported insufficient (low) activity. Self-efficacy positively correlated with age (r = 0.199, *p* < 0.01) and education level (r = 0.191, *p* < 0.01), and negatively with employment (r = –0.171, *p* < 0.05). Physical activity was significantly associated with self-efficacy (r = 0.217, *p* < 0.01). These findings underscore the importance of fostering self-efficacy to promote physical activity, highlighting the need for targeted strategies within academic settings to support student well-being and healthier lifestyle choices.

## 1. Introduction

Currently, one of the most crucial issues we are facing is the potential of an inactive lifestyle to lead to the formation of unhealthy habits. According to the World Health Organization (WHO) [1], non-communicable diseases are the leading cause of premature death in the world. The main risk factors for non-communicable diseases are smoking, excessive alcohol consumption, unhealthy diet, sedentary lifestyle and physical inactivity. Physical inactivity increases the risk of cancer, heart disease, stroke, and diabetes by 20–30% [1]. According to WHO recommendations, adults are encouraged to engage in 150 to 300 min of moderate-intensity aerobic activity per week, or 75 to 150 min of vigorous-intensity aerobic activity, or a comparable mix of both intensities, in order to achieve significant health benefits [2].

The research carried out by the Centre for Disease Prevention and Control [3] on health-affecting habits of the population of Latvia reveals that the proportion of respondents who undergo at least 30 min of physical exercise in their free time until they feel short of breath or sweat every day is 2.9%, 4–6 times per week, it is 2.6%; 2–3 times per week, it is 12.9%; a few times per year, it is 48.9% [3]. According to Special Eurobarometer 525 survey data, 45% of Europeans report they never exercise or play sports [4]. Data from the Special Eurobarometer Survey covering the European Union (EU) sample (N = 26,580) and the Latvian subset (LV, N = 1013) reveal that, within the EU sample, 7% of respondents reported engaging in more than 120 min of vigorous physical activity (PA), 24% reported 60 to 120 min, 48% reported 60 min or less, and 21% reported no vigorous PA or uncertainty. In contrast, the Latvian subset showed higher engagement, with 22% reporting more than 120 min and only 16% reporting no vigorous PA. For moderate PA, in the EU sample, 7% of respondents engaged in over 120 min, with 24% engaging for 60 to 120 min, 59% for 60 min or less, and 10% indicating no engagement or uncertainty. In the Latvian sample, 18% engaged in more than 120 min, showing a higher commitment to moderate PA than the EU average, and 15% reported no moderate PA. In the EU sample, 16% of respondents spent sitting for 150 min or less, 43% for 151 to 330 min, 28% for 331 to 510 min, 11% for 511 min or more, and 2% were uncertain. The Latvian sample reported similar distributions, though with a slight increase in those reporting higher sitting minutes [4].

University students, particularly those studying physiotherapy, face unique challenges in maintaining an active lifestyle due to academic pressures and time constraints [5,6]. As future healthcare professionals are expected to promote physical activity, their own behaviors and confidence in sustaining such behaviors are especially important to understand. However, research indicates that healthcare students often exhibit low levels of physical activity [7]. This study focuses specifically on physiotherapy students to examine the relationship between self-efficacy and their physical activity levels.

Self-efficacy is defined as an individual’s belief in their ability to cope with various life situations [8,9]. A person with low self-efficacy tends to run away or give up when faced with difficulties. In contrast, a person with high self-efficacy will take specific actions to solve a problem when faced with difficulties [10]. Self-efficacy plays a vital role in behavioral self-regulation because it influences the formation and strength of intentions and the persistence of action in the face of obstacles [8,9].

Self-efficacy is widely recognized as a key psychosocial determinant of physical activity, influencing both the motivation to engage and actual exercise behavior [11,12,13,14,15]. It has been shown to correlate positively with moderate-to-vigorous physical activity [16], and can be shaped by perceived barriers such as time constraints, inconvenience, or lack of social support [11]. Additionally, exercise behavior—including intensity, duration, and frequency—can mediate the relationship between physical fitness and self-efficacy, with evidence suggesting that regular participation (e.g., exercising two or more times per week for 30–60 min) is associated with higher self-efficacy levels [17,18].

Although physical activity is an important indicator of health behavior, in a demanding study environment, the scarcity of time [5,6] may lead to the perception that engaging in physical activity is impractical, resulting in lower self-efficacy to incorporate exercise into daily routine. Therefore, the aim of this was to investigate the relationship of self-reported physical activity level and self-efficacy in physiotherapy students. The study analyzes students’ overall self-efficacy levels, physical activity levels, and sedentary behavior duration (sitting minutes), while also examining the relationship of these indicators with demographic characteristics, providing a broader understanding of the interplay between these factors in the context of students’ health behaviors.

## 2. Materials and Methods

A cross-sectional descriptive correlation study was conducted to examine self-efficacy, physical activity, sedentary behavior, and the relationship between self- efficacy and physical activity.

### 2.1. Participants

The research sample consists of 224 physiotherapy 4-year bachelor-level students.

### 2.2. Procedure/Instruments

In total, 250 questionnaires were distributed between December 2023 and April 2024 and 224 complete questionnaires were received and analyzed in this study. The questionnaire consists of 3 parts. The first part included demographic information, such as gender, age, education background, study year and employment. The second part of the questionnaire included the General Self-Efficacy Scale (GSES) [19] and the third part included Physical Activity Questionnaire—Short Form (IPAQ-SF) [20].

GSES is a self-report one-dimensional measure of self-efficacy. It consists of ten statements with four answers. Regarding the internal reliability for GSES, Cronbach’s α is 0.845. The completion of the scale requires four minutes on average. The calculation of the responses uses a four-point Likert scale, assessing a total of 10 statements from 1 to 4 (1 = Not at all true, 2 = Hardly true, 3 = Moderately true, 4 = Exactly true). The total score calculated the sum of all items ranging between 10 and 40, with a higher score indicating more self-efficacy. GSES was chosen in this study based on its extensive use in the student population. GSES is adapted to the Latvian culture and shows good reliability, with Cronbach’s alpha of 0.81, reaction indices from 1.6 to 3.4 and discrimination indices between 0.36 and 0.59 [21].

IPAQ-SF is an internationally recognized tool used in physical activity research [20], and it is adapted to the Latvian language [22]. The questionnaire contains 7 questions that summarize the time spent walking and carrying out moderate-intensity, and vigorous-intensity activities over the past 7 days. MET-min per week were calculated for each activity type by multiplying the reported minutes and days by the corresponding MET value (walking MET-min/week = minutes spent walking × days per week × 3.3 METs, moderate activity MET-min/week = minutes of moderate-intensity activity × days per week × 4.0 METs, vigorous activity MET-min/week = minutes of vigorous-intensity activity × days per week × 8.0 METs) according to the official IPAQ scoring protocol. The total physical activity MET-min/week is the sum of walking, moderate, and vigorous MET-min. Physical activity levels were classified into three categories according to the IPAQ-SF scoring protocol: low—not meeting any criteria for moderate or high activity; moderate—at least 600 MET-min/week, and high—at least 3000 MET-min/week [23].

Many studies [15,17,18,24,25,26,27,28,29,30] used the General Self-Efficacy Scale (GSES) to measure the students’ self-efficacy level proving the methods reliability. IPAQ-SF is an internationally recognized tool used in physical activity research [20]. It has been used in many studies to determine the student physical activity level [15,31,32,33].

### 2.3. Statistical Analysis

For data analysis, we used mathematical statistics with the IBM SPSS 29 program. Descriptive statistical analysis, including frequencies, mean (M), standard deviation (SD), median, minimal and maximal values,, was used to summarize demographic data, self-efficacy scores, and physical activity levels. The Shapiro–Wilk test was used to test data normality for all variables. Spearman’s r was calculated to assess the relationship between demographics, self-efficacy and physical activity level. Statistical significance was assessed using a two-tailed *p*-value with an alpha level set at 0.05. Cronbach’s alpha for GSES was calculated showing good reliability 0.84.

## 3. Results

A total of 224 respondents (44 males and 180 females) completed and returned questionnaires, which were analyzed in this study. Their mean age was 24.44 (SD 7.56). Socio-demographic data were collected for each participant, including age, gender, education level, study year and employment. The characteristics of participants are shown in Table 1. Th students’ previous education levels with diverse backgrounds included general secondary education, incomplete higher education, and general professional or higher education. Participants included students from various years of study, with the majority representing the second year. Employment status varied, with 59.4% of students in either full-time or part-time employment, and 40.6% reporting as unemployed.

The results of the IPAQ questionnaire were analyzed and PA levels were calculated. In total, 149 out of 224 participants reported their PA level. A total of 60 participants (40.3%) achieved a high level of physical activity, exceeding 3000 METs per week. The largest group, comprising 77 participants (51.7%), engaged in a moderate level of physical activity, with METs ranging between 600 and 3000 per week. Only 12 participants (8.1%) reported low physical activity, with METs at or below 600 per week [22].

Data on physical activity and sedentary behavior were categorized based on responses to the International Physical Activity Questionnaire—Short Form (IPAQ-SF), in which participants reported the number of minutes they engaged in physical activity on days they performed it. The results were grouped into four categories for each activity domain: vigorous physical activity, moderate physical activity, and time spent sitting. For vigorous physical activity, 4.5% of respondents reported engaging in more than 120 min per week, 13.4% reported between 61 and 120 min, and 17.0% reported 60 min or less. The majority—65.2%—indicated they either never performed vigorous activity or were unsure of their activity levels. For moderate physical activity, 9.4% of respondents reported more than 120 min per week, 12.1% reported between 61 and 120 min, and 28.6% engaged in 60 min or less. Half of the respondents (50.0%) reported no moderate activity or were unsure of their participation. Regarding daily sitting time, 4.9% of participants reported sitting for 150 min or less, 26.8% sat between 151 and 330 min, and 24.6% sat between 331 and 550 min daily. A further 39.7% (n = 89) reported sitting for 551 min or more per day, while 4.0% (n = 9) did not provide an estimate.

In addition, participants’ weekly physical activity levels were converted into MET-min to quantify energy expenditure across different activity intensities. For vigorous physical activity (n = 78), the mean was 2291.00 MET-min per week (SD = 1971.00), with a median of 1920.00, ranging from 120.00 to 9600.00 MET-min/week. Moderate physical activity (n = 112) showed a mean of 1341 MET-min per week (SD = 1269.00), a median of 960.00, and a range from 60 to 7200. Walking activity (n = 139) had a mean of 1565.00 MET-min per week (SD = 2376.00), a median of 808.00, and ranged from 99.00 to 22,275.00 MET-min/week. These results indicate considerable variation in the physical activity patterns among physiotherapy students, with some individuals reporting high levels of activity, while others were notably less active.

Measured among 224 participants, self-efficacy scores ranged from 17.00 to 40.00, with a mean of 30.44 (SD = 3.93). Among 149 participants, total PA MET per week varied widely, with MET values ranging from 99.00 to 24,075.00. The mean physical activity level was 3673.00 METs (SD = 3677.00). For the 135 participants who reported their sitting time, daily sitting minutes ranged from 20.00 to 840.00, with an average of 334.00 (SD = 150.00) (Table 2).

Spearman correlation analyses were used for self-efficacy, PA and total sitting minutes and demographic variables (Table 3). PA shows statistically significant positive correlation with self-efficacy (r = 0.217). Self-efficacy shows significant positive correlation with age (r = 0.199), education level (r = 0.191) and negative correlation with employment (r = −0.171). In addition, total sitting minutes show significant positive correlation with employment (r = 0.180) and negative correlation with study year (r = −0.201).

## 4. Discussion

This study highlights the significant relationship between self-efficacy and physical activity levels among physiotherapy students, emphasizing the importance of psychological factors in promoting active lifestyles in this population.

In terms of PA, 40.3% of students reported a high level, 51.7% reported a moderate level (meeting minimum guidelines), and 8.05% reported an insufficient (low) level of activity. These findings are consistent with previous research showing low and moderate physical activity among medical students [34]. In contrast, other studies show that healthcare students perform enough PA [7]. Comparing current study data with EU sample and LV subsample data from the Special Eurobarometer Survey [4], physiotherapy students showed lower engagement in vigorous and moderate PA relative to both the broader EU and Latvian samples. In the current study sample (N = 224), 4% of respondents reported over 120 min of vigorous PA compared to the EU sample (7%) and LV sample (22%), while the majority (65%) reported no vigorous PA or uncertainty. Moderate PA levels were lower in the current study sample compared to the LV sample (18%) and slightly higher than the EU sample (7%), with only 9% reporting more than 120 min, while the majority (51%) reported no moderate PA or uncertainty, contrasting with the EU and Latvian engagement levels. In the current study sample, 40% of participants were uncertain about sitting minutes. In addition, 4% of respondents reported sitting for 150 min or less, 27% reported between 151 and 330 min and 22% reported between 131 and 150 min, with only 6% reporting 511 and more minutes spent sitting, showing lower siting behavior relative to the EU and Latvian samples [4].

Consistent with previous research, self-efficacy demonstrated a key positive correlation with physical activity, reinforcing its role as a crucial determinant of exercise behavior [26,28,35]. The positive association between self-efficacy and factors such as age and education level may reflect the cumulative impact of experience and mastery, as proposed by Bandura’s theory. Mastery experience is the most influential source of self-efficacy. It refers to the subjective experiences of success or failure that individuals have when they attempt to perform a task or achieve a goal [8,9]. Conversely, the negative correlation between employment and self-efficacy suggests that external demands may reduce students’ confidence in maintaining physical activity, highlighting the need for support mechanisms for working students. Students who were not employed or were part time employed showed higher self-efficacy. This could be because students can focus solely on their studies, thus showing a higher level of confidence. These findings are consistent with research that concluded that older students (over 31.5 years) are likely to have higher academic performance (grades). This could be explained by the maturity of these students who have established study patterns, time management skills, and clear motivation toward their studies [36]. Additionally, students who engage in strength training and physical activity in general show positive effects on academic performance [37].

The elevated sedentary behavior among employed students and those in later years of study aligns with existing evidence that academic and work pressures often limit opportunities for physical activity [5,6]. Studies show that inactivity and increased sedentary behavior correlates with psychological health, elevating stress, anxiety, and depression [38]. Given the well-established links between sedentary lifestyles and adverse mental health outcomes, these findings underscore the importance of integrating physical activity promotion and self-efficacy enhancement strategies into the educational environment.

The study’s findings are subject to certain limitations that should be considered when interpreting the results. First, the sample comprised exclusively physiotherapy students, which may restrict the generalizability of the findings to other student populations. Second, the use of the IPAQ-SF, while widely recognized, provides only estimations of physical activity and may fail to capture finer details such as variations in intensity or unreported activities. Finally, the reliance on self-reported measures for both physical activity and self-efficacy introduces potential biases, including inaccuracies stemming from subjective perceptions and recall errors by participants. These limitations highlight the need for cautious extrapolation of the results and suggest directions for future research employing more diverse samples and objective measurement methods.

## 5. Conclusions

In conclusion, this study provides valuable insights into the relationship between self-efficacy, physical activity, and sitting time among students, while taking into consideration demographic factors. While students exhibit moderate to high self-efficacy, there is a need to address physical inactivity and sedentary behavior, particularly for students with work or academic commitments. The findings underscore the importance of considering psychological factors, such as self-efficacy, when designing interventions to promote physical activity and overall health among student populations. There is a need for future research that focuses on evaluating the effectiveness of psychological support and motivational programs in facilitating the integration of physical activity into students’ daily routines and assessing their long-term impact on well-being and academic performance.

## Figures and Tables

**Table 1 ijerph-22-01029-t001:** Characteristics of participating students in the survey.

Participant Characteristics	Frequency	%
Education background		
General secondary	106	47.3
General professional	13	5.8
Incomplete higher	62	27.7
Higher	43	19.2
Study year		
1st year	57	25.4
2nd year	90	40.2
3rd year	51	22.8
4th year	26	11.6
Employment		
Full time	47	21
Part time	86	38.4
Unemployed	91	40.6
PA level		
High PA (>3000 METs)	60	40.30
Moderate PA (>600; ≤3000 METs)	77	51.70
Low PA (≤600 METs)	12	8.10
Vigorous PA minutes		
More than 120 min	10	4.5
61 to 120 min	30	13.4
60 min and less	38	17.0
Never/Do not know	146	65.2
Moderate PA minutes		
More than 120 min	21	9.4
61 to 120 min	27	12.1
60 min and less	64	28.6
Never/Do not know	112	50.0
Time spent sitting		
150 min or less	11	4.9
151 to 330 min	60	26.8
331 to 550 min	55	24.6
551 min and more	9	4.0
Do not know	89	9.7

**Table 2 ijerph-22-01029-t002:** Descriptive statistics of physical activity, total sitting minutes, and self-efficacy.

	N	Mean	SD	Median	Min	Max
Vigorous PA MET (per week)	78	2291	1971	1920	120	9600
Moderate PA MET (per week)	112	1341	1269	960	60	7200
Walking MET (per week)	139	1565	2376	808	99	22,275
Total PA MET (per week)	149	3673	3677	2772	99	24,075
Total sitting minutes (per day)	135	334	150	300	20	840
Self-efficacy	224	30.44	3.94	30.00	17.00	40.00

**Table 3 ijerph-22-01029-t003:** Correlation of self-efficacy, physical activity and total sitting minutes.

		Self-Efficacy	Physical Activity (METs Per Week)	Total Sitting Minutes
Self-efficacy	Spearman Correlation	1	0.217 **	−0.072
	Sig. (2-tailed)		0.008	0.411
	N	224	149	134
Age	Spearman Correlation	0.199 **	0.098	−0.118
	Sig. (2-tailed)	0.003	0.235	0.173
	N	224	149	134
Education background	Spearman Correlation	0.191 **	0.147	−0.116
	Sig. (2-tailed)	0.004	0.074	0.056
	N	224	149	134
Study year	Spearman Correlation	0.010	−0.011	−0.201 *
	Sig. (2-tailed)	0.879	0.898	0.020
	N	224	149	134
Employment	Spearman Correlation	−0.171 *	−0.068	0.180 *
	Sig. (2-tailed)	0.010	0.407	0.038
	N	224	149	134

* Correlation is significant at the 0.05 level (2-tailed). ** Correlation is significant at the 0.01 level (2-tailed).

## Data Availability

The data are available from the Riga Stradins University Dataverse database upon request at https://doi.org/10.48510/FK2/TMBSSQ.

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
