# Peer review of "The Relationship of Self-Reported Physical Activity Level and Self-Efficacy in Physiotherapy Students: A Cross-Sectional Study"

_ijerph, 2025, doi:10.3390/ijerph22071029_

Round 1

Reviewer 1 Report

Comments and Suggestions for Authors

Review IJEPRH

General
This is a well-conducted small study with important findings that should be shared with the public. It is well written. However, there are several issues that should be addressed before acceptance.

Abstract
Well written and clear.

Introduction

  1. Line 27 – If you are quoting the WHO, you need to add it to the references.
  2. Line 44 – I am not sure Bandura is relevant here. Consider shortening this sentence.
  3. Lines 44–50 discuss self-efficacy, but this concept has already been addressed in earlier lines. I recommend shortening this section and moving it before lines 39–43. Then, the paragraph from lines 39–43 should be combined and shortened with lines 51–60, as they are repetitive.
  4. In the last paragraph of the introduction, you refer to the students who participated in the study. This is the first time the population is mentioned, yet it is a crucial part of the study. You should introduce and justify the choice of this population earlier in the introduction. This needs to be emphasized more clearly.

Methods

  1. Section 2.1, line 75 – You state that the sample size is 244, but in section 2.2 you mention 224. Which is correct?
  2. In the results, you present findings using METs, but this is not mentioned anywhere in the methods. You need to explain how METs were used in the context of the physical activity data collected (i.e., how you converted the measured physical activity into METs).

Results

  1. In Table 1, please clarify the education section. How do the participants have varying education levels if they are all physiotherapy students?
  2. Line 130 and Table 3 – You use METs, yet METs were not mentioned earlier in the manuscript and were not described as a measurement tool in the methods. Either refer to the original physical activity measures used in the questionnaire or explain how the data were converted into METs.
  3. Lines 136–159 and Table 4 – You compare your results to data that are not part of your study. This should not be included in the results section; instead, present your results here and move the comparison to the discussion section.
  4. Table 4 – The title is unclear.
  5. Table 6 – You cannot use METs unless you have explained them clearly in the methods section.
  6. In Table 6, the educational status is unclear. Aren’t all participants physiotherapy students?

Discussion

  1. Line 181 – You report a mean age of 20.6, but in the results section, it is 24.44. Which is correct?
  2. The same issue regarding METs applies throughout the manuscript.

References

There are way too many references from the same journal (11/37): Front Psychol.

Author Response

General
Comment 1: This is a well-conducted small study with important findings that should be shared with the public. It is well written. However, there are several issues that should be addressed before acceptance.

Response 1: We sincerely thank the reviewer for the positive and encouraging feedback on the importance and clarity of our study. We appreciate the recognition of its potential contribution to the literature. Below, we address all specific concerns and suggestions in detail, and we have revised the manuscript accordingly.

Abstract
Comment 2: Well written and clear.

Response 2: We thank the reviewer for the kind comment regarding the abstract. We are pleased that the abstract was found to be clear and well written.

Introduction

Comment 3: Line 27 – If you are quoting the WHO, you need to add it to the references.

Response 3: Thank you for pointing this out. We have now added the appropriate citation to the World Health Organization (WHO) source in the reference list.

Comment 4: Line 44 – I am not sure Bandura is relevant here. Consider shortening this sentence.

Response 4: We appreciate the reviewer’s suggestion. We have revised the sentence to be more concise and focused, removing the specific reference to Bandura while retaining the conceptual link to self-efficacy.

Comment 5: Lines 44–50 discuss self-efficacy, but this concept has already been addressed in earlier lines. I recommend shortening this section and moving it before lines 39–43. Then, the paragraph from lines 39–43 should be combined and shortened with lines 51–60, as they are repetitive.

Response 5: Thank you for this valuable feedback. We acknowledge the repetition in the discussion of self-efficacy and appreciate the suggestion to improve the structure and flow. We have moved the condensed discussion of self-efficacy earlier in the paragraph, before lines 39–43, and combined and revised the content from lines 39–43 and 51–60 to eliminate repetition. The changes are now visible from lines 64-77.

Comment 6: In the last paragraph of the introduction, you refer to the students who participated in the study. This is the first time the population is mentioned, yet it is a crucial part of the study. You should introduce and justify the choice of this population earlier in the introduction. This needs to be emphasized more clearly.

Response 6: Thank you for this helpful suggestion. We agree that the study population - physiotherapy students - should be introduced earlier in the introduction and its relevance more clearly explained. We have now added the second paragraph from lines 39-44 to include an earlier mention of the study population and to justify the focus on this group, based on their academic demands and future role as health promoters.

Methods

Comment 7: Section 2.1, line 75 – You state that the sample size is 244, but in section 2.2 you mention 224. Which is correct?

Response 7: Thank you for identifying this discrepancy. The correct sample size is 244 participants. The mention of 224 in Section 2.2 was an error, which has now been corrected in the revised manuscript to ensure consistency across all sections.

Comment 8: In the results, you present findings using METs, but this is not mentioned anywhere in the methods. You need to explain how METs were used in the context of the physical activity data collected (i.e., how you converted the measured physical activity into METs).

Response 8: Thank you for your observation. The explanation of MET calculations based on the IPAQ-SF was originally included in the methodology; however, we understand that this may not have been sufficiently clear. To address this, we have revised the relevant section in the Methods to improve clarity and ensure the role of METs is explicitly described from lines 112-122.

Results

Comment 9: In Table 1, please clarify the education section. How do the participants have varying education levels if they are all physiotherapy students?

Response 9: Thank you for pointing this out. We agree that the education section may have been misleading. In this study, “education” refers to participants’ educational background prior to enrolling in the bachelor-level physiotherapy program. This includes general secondary education, incomplete higher education, and completed higher or professional education. We have revised Table 1 and the relevant text to clarify this definition and avoid confusion.

Comment 10: Line 130 and Table 3 – You use METs, yet METs were not mentioned earlier in the manuscript and were not described as a measurement tool in the methods. Either refer to the original physical activity measures used in the questionnaire or explain how the data were converted into METs.

Response 10: Thank you for raising this issue. We believe that now the calculation of METs is explained earlier in the methods form line 112-122.

Comment 11: Lines 136–159 and Table 4 – You compare your results to data that are not part of your study. This should not be included in the results section; instead, present your results here and move the comparison to the discussion section.

Response 11: Thank you for this helpful structural suggestion. In response, we have made the following changes: 1) Information about the SES study is now presented in the Introduction; 2) Our findings are presented in the Table 1; 3)All comparisons between our findings and the SES data are now located exclusively in the Discussion.

Comment 12: Table 4 – The title is unclear.

Response 12: Thank you for pointing this out. We have removed Table 4 and incorporated our findings in Table 1.

Comment 13: Table 6 – You cannot use METs unless you have explained them clearly in the methods section.

Response 13: We believe that now the calculation of METs is explained earlier in the methods form line 90-98.

Comment 14: In Table 6, the educational status is unclear. Aren’t all participants physiotherapy students?

Response 14: We agree that the term “education level” may have been misleading. In this study, “education level” refers to participants’ educational background prior to enrolling in the bachelor-level physiotherapy program. We have revised Table 6 and the relevant text to clarify this definition and avoid confusion.

Discussion

Comment 15: Line 181 – You report a mean age of 20.6, but in the results section, it is 24.44. Which is correct?

Response 15: Thank you for noticing this discrepancy. The correct mean age of the participants is 24.44 years (±7.56) as reported in the Results section. The value of 20.6 stated in Line 181 was an error, and we have now corrected it to reflect the accurate data.

Comment 16: The same issue regarding METs applies throughout the manuscript.

Response 16: Thank you for noticing this discrepancy. We have now corrected it to reflect the accurate data in lines from 163-364, line 189.

References

Comment 17: There are way too many references from the same journal (11/37): Front Psychol.

Response 17: Thank you for your observation. We recognize the importance of ensuring diversity and balance in cited sources. However, many relevant studies were published in Frontiers in Psychology due to its focus on health and behavioral science.

Reviewer 2 Report

Comments and Suggestions for Authors

I would firstly like to thank the authors for their efforts in putting together this manuscript.  The topic of the manuscript has the potential to provide novel insight into the physical behaviours and capacity for physical activity in physiotherapy students. This topic has the potential to be very relevant given the increase in strain on health services, in addition to the disconnect between working in the health sector and having the support / correct environment to be physically active. It is also relevant when considering the position of university students on development and decision making on behaviours.

However, despite this, in its current form the manuscript required significant amendments in order to be considered for publication. Detail is needed in the methods which clearly outline how the data is processed, which variables are generate, which are included in the analysis and how the significance of the findings are assessed. I would consider the variables used and how best to present these in relation to the literature with which they will subsequently be compared to. The results should be more concise and focus on the findings of this study rather than comparing to other studies. The discussion should be more concise and less descriptive, focus should be placed here on the implications of the study findings and its position within the literature rather than describe the already presented findings.

In addition, to these above general points, see below for more specific suggestions which would benefit from clarity / consideration: 

Abstract

  • Line 18: Please clarify what ‘Moderate’ refers to in this context. It is assumed that sufficient and insufficient refer, to meeting the guidelines vs not meeting the guidelines, therefore is moderate referring to a intermediate threshold? If so, please define this.
  • Line 19: The correlations presented, are quantified as weak correlations (based on the r squared value), please clarify if these values statistically significant.
  • Line 20: The association is stated as significant, please include detail of this.

Introduction

  • Line 28: WHO is directly referred to but is not cited. Please include a reference.

Methods

  • Line 98: Was ‘Total physical activity per week’ is the only variable mention as being generated from the IPAQ. Please detail all variables used, there are variables presented in the results section which are not detailed, with regards to how they are generated.
  • Section 2.3: More detail on the analysis is needed in this section. Please detail which variables will be used / compared in each analysis.
  • Line 109: Please include detail of how significance was assessed and the alpha level used to assess it.

Results

I would suggest simplifying the data presented across in multiple tables and consolidating  this for clarity into a descriptive table and an analysis table. 

  • Table 2: In its current form Table 2 is not very intuitive. Typically physical activity is referred to in relation to a guideline or normative value relative to a weekly value (such as 150 minutes MVPA/week  or 3,000 METs/ week). As such I would suggest considering how the data is presented in this table to align to that rather than presenting the number of days it was completed. For example stating the amount of Moderate PA completed per week etc.
  • Table 2: More clarity is needed in the description of the variables presented. I assume the ‘Days of vigorous PA (per week)’ refers to the number of days in which VPA was undertaken in some amount, however, as it is written this could be interpreted that on average participants completed 2.96 days worth of VPA a week, which is different to that being the number of days in which some VPA was undertaken. I would consider revising these descriptions, (this may be covered if the table is revised base on my previous comment).
  • Line 130: More clarity is needed here, please clarify if the time period these normative values presented related to. Whilst is it may be assumed this is per week this should be stated. In addition, please include a reference for where these normative values have come from.
  • Table 3: The detail provided in this table relating to normative METs values should be defined first in your methods, before being presented here. Please include detail to this and all variables used in the analysis in the methods section.
  • Lines 136-159: Comparison is made to the SES, which presents data in minutes of VPA or MPA. However the data presented in this study is presented in METs Per week. Despite this information from this study is directly compared. Please present the data being used to make this comparison.
  • Comparison to SES: Data from existing studies should be presented in the introduction, with comparisons to the results of this study presented in the discussion. Further all existing data should be reference appropriately.
  • Table 4: The data presented in Table 4 is not directly referred to or cited in text. Please include a citation to this table.
  • Lines 136- 159 / Table 4: Comparisons made to existing data can only be made on a descriptive level unless statistical analysis is carried out to support the comparison.

Discussion

  • Line 197: The data presented here is not presented in the results. For clarity please include all data presented  in the discussion in the results sections. void presenting information in the discussion for the first time. 

Author Response

Comment 1: I would firstly like to thank the authors for their efforts in putting together this manuscript.  The topic of the manuscript has the potential to provide novel insight into the physical behaviours and capacity for physical activity in physiotherapy students. This topic has the potential to be very relevant given the increase in strain on health services, in addition to the disconnect between working in the health sector and having the support / correct environment to be physically active. It is also relevant when considering the position of university students on development and decision making on behaviours.

 Response 2: We sincerely thank the reviewer for these encouraging and thoughtful comments. We appreciate your recognition of the relevance of this topic, especially in light of the evolving demands on healthcare professionals and the importance of promoting healthy behaviors early in their academic and professional journey.

Comment 2: However, despite this, in its current form the manuscript required significant amendments in order to be considered for publication. Detail is needed in the methods which clearly outline how the data is processed, which variables are generate, which are included in the analysis and how the significance of the findings are assessed. I would consider the variables used and how best to present these in relation to the literature with which they will subsequently be compared to. The results should be more concise and focus on the findings of this study rather than comparing to other studies. The discussion should be more concise and less descriptive, focus should be placed here on the implications of the study findings and its position within the literature rather than describe the already presented findings.

 Response 2: We thank the reviewer for this thorough and insightful feedback, which we found very helpful in strengthening the quality of our manuscript. In response, we have made the following major revisions to address the concerns raised:

1)Methods Section:

  • We have expanded the Methods section to clearly describe how raw data were processed, how physical activity levels were converted into MET-minutes/week based on IPAQ-SF scoring guidelines.
  • All variables included in the analysis are now explicitly listed and defined, including sociodemographic factors, General Self-Efficacy Scale scores, and physical activity categories (low, moderate, high).
  • We have also provided more detail on the statistical methods used, including the rationale for using Spearman’s correlation coefficients, descriptive statistics, and significance thresholds (p < 0.05).

2)Results Section:

  • The results have been revised for conciseness and focus. We now present the main findings more clearly and have minimized unnecessary comparisons to prior studies in this section.
  • Comparative interpretation with prior literature has been moved to the Discussion section, where it is more appropriate.

3)Discussion Section:

  • The Discussion has been rewritten to be less descriptive and more analytical.
  • We have minimized repetition of the Results and instead focused on the implications of the findings, particularly the role of self-efficacy in shaping physical activity behaviors among physiotherapy students.
  • We have also critically discussed how our findings relate to existing literature and what this may mean for interventions targeting physical activity and psychosocial support in health-related university programs.

Abstract

Comment 3: Line 18: Please clarify what ‘Moderate’ refers to in this context. It is assumed that sufficient and insufficient refer, to meeting the guidelines vs not meeting the guidelines, therefore is moderate referring to a intermediate threshold? If so, please define this.

Response 3: Thank you for highlighting this point. We agree that the use of the term “moderate” required clarification. In this study, physical activity levels were classified according to the IPAQ-SF scoring protocol, which defines three categories: low (insufficient activity), moderate, and high (sufficient activity) based on total MET-minutes/week. To clarify, low PA means it does not meet the criteria for moderate or high activity (i.e., insufficient activity), moderate PA means it meets the minimum recommended level of physical activity — e.g., 600 MET-minutes/week — but not the higher thresholds, and high PA (Sufficient) means it exceeds the minimum recommendations, typically >3,000 MET-minutes/week or equivalent combinations of intensity and duration. We have revised Line 18 and the Methods section from line 99-103 to clearly define these thresholds in alignment with IPAQ guidelines.

Comment 4: Line 19: The correlations presented, are quantified as weak correlations (based on the r squared value), please clarify if these values statistically significant.

Response 4: Thank you for this observation. You are correct that the reported correlation coefficients (r values) represent weak associations in terms of effect size. However, we confirm that all the correlations reported — including those between self-efficacy and age (r = 0.199), education level (r = 0.191), employment (r = –0.171), and physical activity (r = 0.217) — were statistically significant with p-values < 0.05. We have revised Lines from 17-22 in the Abstract and clarified the significance of these correlations in the Results section to explicitly state which correlations were statistically significant.

Comment 5: Line 20: The association is stated as significant, please include detail of this.

Response 5: Please see the response 4.

Introduction

Comment 6: Line 28: WHO is directly referred to but is not cited. Please include a reference.

Response 6: Thank you for pointing this out. We have now added the appropriate citation to the World Health Organization (WHO) source in the reference list.

Methods

Comment 7: Line 98: Was ‘Total physical activity per week’ is the only variable mention as being generated from the IPAQ. Please detail all variables used, there are variables presented in the results section which are not detailed, with regards to how they are generated.

Response 7: Thank you for this important comment. We agree that the Methods section initially lacked sufficient detail regarding the full set of variables generated from the IPAQ-SF. In response, we have revised the Methods to include a full description of all derived variables and the specific calculations used from lines 94-104.

Comment 8: Section 2.3: More detail on the analysis is needed in this section. Please detail which variables will be used / compared in each analysis.

Response 8: Thank you for pointing out this important omission. We have revised Section 2.3 to clearly specify which variables were included in each statistical analysis, and how relationships between them were assessed. This should provide greater transparency and replicability for readers.

Comment 9: Line 109: Please include detail of how significance was assessed and the alpha level used to assess it.

Response 9: Thank you for your comment. We have revised the manuscript to clarify how statistical significance was assessed. Specifically, we have stated the alpha level used (p < 0.05) and have clarified that all tests were two-tailed.

Results

Comment 10: I would suggest simplifying the data presented across in multiple tables and consolidating  this for clarity into a descriptive table and an analysis table. 

Response 10: Thank you for your valuable suggestion. We have simplified the presentation of our data by consolidating the information into three main tables: two descriptive tables summarizing participant characteristics and physical activity measures, and one correlation analysis table presenting the key statistical findings. This restructuring improves clarity and makes the results easier to interpret.

Comment 11: Table 2: In its current form Table 2 is not very intuitive. Typically physical activity is referred to in relation to a guideline or normative value relative to a weekly value (such as 150 minutes MVPA/week  or 3,000 METs/ week). As such I would suggest considering how the data is presented in this table to align to that rather than presenting the number of days it was completed. For example stating the amount of Moderate PA completed per week etc.

Response 11: Thank you for your insightful feedback. In response, we have revised Table 2 to align physical activity data with established guidelines by presenting the amount of moderate and vigorous physical activity completed per week, rather than the number of days. This approach better reflects standard metrics such as minutes per week or MET-minutes per week.

Comment 12: Table 2: More clarity is needed in the description of the variables presented. I assume the ‘Days of vigorous PA (per week)’ refers to the number of days in which VPA was undertaken in some amount, however, as it is written this could be interpreted that on average participants completed 2.96 days worth of VPA a week, which is different to that being the number of days in which some VPA was undertaken. I would consider revising these descriptions, (this may be covered if the table is revised base on my previous comment).

Response 12: Pleasse see the response 11.

Comment 13: Line 130: More clarity is needed here, please clarify if the time period these normative values presented related to. Whilst is it may be assumed this is per week this should be stated. In addition, please include a reference for where these normative values have come from.

Response 14: Thank you for this helpful comment. We agree that the time frame for the normative values should be clearly stated and supported by a citation. We have revised Line 130 to specify that these values refer to weekly physical activity levels and added an appropriate reference.

Comment 15: Table 3: The detail provided in this table relating to normative METs values should be defined first in your methods, before being presented here. Please include detail to this and all variables used in the analysis in the methods section.

Response 15: Thank you for this important observation. We agree that all variables used in the analysis, including how MET values and normative categories were derived, should be clearly defined in the Methods section.

Comment 16: Lines 136-159: Comparison is made to the SES, which presents data in minutes of VPA or MPA. However the data presented in this study is presented in METs Per week. Despite this information from this study is directly compared. Please present the data being used to make this comparison.

Response 16: Thank you for this thoughtful observation. You are correct that the comparison originally presented between our study results (in MET-minutes/week) and the SES data (in minutes/week of VPA/MPA) was not clearly aligned. We now present both raw minutes/week (Table 1)  and MET-minutes/week (Table 2)and (derived from IPAQ) where available in our dataset.

Comment 17: Comparison to SES: Data from existing studies should be presented in the introduction, with comparisons to the results of this study presented in the discussion. Further all existing data should be reference appropriately.

Response 17: Thank you for this helpful structural suggestion. In response, we have made the following changes: 1) Information about the SES study is now presented in the Introduction; 2) All comparisons between our findings and the SES data are now located exclusively in the Discussion.

Comment 18: Table 4: The data presented in Table 4 is not directly referred to or cited in text. Please include a citation to this table.

Response 18: Thank you for this observation. In response to Comment 10, we have revised the structure of our results section and consolidated multiple tables into a descriptive table and an analysis table for improved clarity and organization. As a result, the content previously included in Table 4 has now been incorporated into the new analysis table (Table 1).

Comment 19: Lines 136- 159 / Table 4: Comparisons made to existing data can only be made on a descriptive level unless statistical analysis is carried out to support the comparison.

Response 19: Thank you for this important clarification. We agree that without formal statistical testing, comparisons with existing studies such as SES should be presented only descriptively. In response, we have revised lines 136–159 and added findings from SES to the Introduction and descriptive comparison located in the Discussion.

Discussion

Comment 20: Line 197: The data presented here is not presented in the results. For clarity please include all data presented  in the discussion in the results sections. Void presenting information in the discussion for the first time. 

Response 20: Thank you for your helpful comment. We have revised the manuscript to ensure that all data referenced in the discussion are now fully presented and described in the results section

Round 2

Reviewer 1 Report

Comments and Suggestions for Authors

The authors have addressed and properly corrected all comments and critiques, except for one issue:

In the Discussion the authors still report a mean age of 20.6, although their answer was:

“Response 15: Thank you for noticing this discrepancy. The correct mean age of the participants is 24.44 years (±7.56) as reported in the Results section. The value of 20.6 stated in Line 181 was an error, and we have now corrected it to reflect the accurate data.”

This was not corrected.

Author Response

Comment 1: 

The authors have addressed and properly corrected all comments and critiques, except for one issue:

In the Discussion the authors still report a mean age of 20.6, although their answer was:

“Response 15: Thank you for noticing this discrepancy. The correct mean age of the participants is 24.44 years (±7.56) as reported in the Results section. The value of 20.6 stated in Line 181 was an error, and we have now corrected it to reflect the accurate data.”

This was not corrected.

Response 1: Thank you for drawing our attention to this. We sincerely apologize for the confusion. While we had addressed this issue in our previous response and corrected the mean age to 24.44 years (±7.56) in the revised manuscript, it appears that this change was either overlooked during the review process or not reflected in the version consulted. We confirm that the incorrect value of 20.6 has been removed in the Discussion section to ensure consistency with the Results section.

Reviewer 2 Report

Comments and Suggestions for Authors

I would like to thank the authors for the time taken to revise the manuscript and address the points raised in the initial review. I feel the authors have sufficiently addressed the points raised the current version of the manuscript is much improved from the initial submission. 

Author Response

Comment 1: I would like to thank the authors for the time taken to revise the manuscript and address the points raised in the initial review. I feel the authors have sufficiently addressed the points raised the current version of the manuscript is much improved from the initial submission. 

Response 2: We sincerely thank you for your thoughtful and constructive feedback throughout the review process. We truly appreciate your kind words and are glad to hear that you find the current version of the manuscript significantly improved. Your comments greatly contributed to enhancing the clarity and quality of our work.